# On the asymptotic behavior of the average geodesic distance $L$ and the compactness $C_B$ of simple connected undirected graphs whose order approaches infinity

Tatiana Lokot[1]☯, Olga Abramov[2]☯*, Alexander Mehler[3]☯

**1** Retired, Faculty of Mathematics, Bielefeld University, Bielefeld, Germany, **2** Affiliation CITEC/Faculty of Technology, Bielefeld University, Bielefeld, Germany, **3** Affiliation Faculty of Computer Science and Mathematics, Goethe-University Frankfurt, Frankfurt, Germany

☯ These authors contributed equally to this work.
* oabramov@techfak.uni-bielefeld.de

**Data Availability Statement:** All relevant data are within the manuscript.

**Funding:** The authors received no specific funding for this work.

## Abstract

The average geodesic distance $L$ Newman (2003) and the compactness $C_B$ Botafogo (1992) are important graph indices in applications of complex network theory to real-world problems. Here, for simple connected undirected graphs $G$ of order $n$, we study the behavior of $L(G)$ and $C_B(G)$, subject to the condition that their order $|V(G)|$ approaches infinity. We prove that the limit of $L(G)/n$ and $C_B(G)$ lies within the interval $[0;1/3]$ and $[2/3;1]$, respectively. Moreover, for any not necessarily rational number $\beta \in [0;1/3]$ ($\alpha \in [2/3;1]$) we show how to construct the sequence of graphs $\{G\}$, $|V(G)| = n \to \infty$, for which the limit of $L(G)/n$ ($C_B(G)$) is exactly $\beta$ ($\alpha$) (Theorems 1 and 2). Based on these results, our work points to novel classification possibilities of graphs at the node level as well as to the information-theoretic classification of the structural complexity of graph indices.

## Introduction

The average geodesic distance is one of the most important graph indices in applications of complex network theory [1–7]. According to the definition of graphs, there are hardly any graph indices that do not rely on transitive dependencies between indirectly connected nodes and thus on a building block of networking information that is essential for complex networks and the processes that run on them [8]. In a survey paper [9], discuss 13 studies of clinical conditions investigated using so-called clinical cognitive networks. In nine of these studies, the patient groups had significantly different average geodesic distance values of their (group-wise aggregated) lexical networks compared to the control groups. This level of informativeness was not achieved by any other of the graph indices examined in these studies. In particular, geodesic distances are analyzed to capture dependencies, over short as well as long distances, to complement local information captured by cluster values and degree statistics (see [1] for an

**Competing interests:** The authors have declared that no competing interests exist.

introduction; see [10] for the representational concept of indirect connections of nodes in cognitive networks).

This distance information is also at the core of a variety of graph entropy measures that characterize nodes via probability distributions based on distances to nodes in their neighborhoods (see [11] for an overview of this research). In addition, node distance statistics play a prominent role in computing prominence, significance, influence, or other aspects of importance of nodes in information networks (for example [12], examine, among other things, the special role of shortest path statistics and related measures such as diameter and eccentricity in research on Twitter), geospatial networks [13–15], or brain networks [5, 16, 17]. These examples point to the central relational information quality of geodesic distance, which is at the center of the relational definition of entities by reference to their neighborhoods, especially indirect ones, as first elaborated in structuralism for a number of sciences. To put it the other way around, relational research approaches, whether in computational sociology [18, 19], linguistics [20–23], chemistry [24–27], or biology [3, 28–30], to name just a few examples, make network statistics based on the evaluation of shortest paths, such as those evaluated by the average geodesic distance, indispensable.

In so far as geodesic distance concerns connected vertices, the question arises on how to additionally capture pairs of disconnected vertices in a network. One possible answer, that was first given in the context of web research, is the compactness measure [31]. By this measure, disconnected nodes and disconnected subgraphs contribute to the compactness value of a network as do connected nodes. This approach eliminates the need to focus on, for example, the largest connected component of a graph. There is a number of studies applying the compactness measure of [31] to answer questions concerning the structure of hypermedia or web-based systems in general [20, 31–37]. What these applications have in common is that they rely on the empirical calculation of compactness values rather than theoretically describing and determining the boundary conditions of their possible manifestations. In this way, they bypass a theoretical description of the functional relationship between average geodesic distance and compactness. In this paper, we focus on this research gap. In our previous paper [38], we conjectured that the limit value of compactness for any sequence of simple connected undirected graphs lies within the interval $\left[\frac{2}{3}; 1\right]$ and for any number $\alpha$ in the interval $\left[\frac{2}{3}; 1\right]$ one can construct the sequence of graphs for which the limit value of compactness is exactly $\alpha$. In the present paper, we prove these two statements about compactness as well as related statements about the asymptotic behaviour of the average geodesic distance (Theorems 1 and 2). In this way, we demonstrate an alternative to the prevailing *practical* evaluation of graph indices in complex network theory by *theoretically* elaborating on the functional relationship between average geodesic distance and one of its relatives, that is, compactness. As outlined in the discussion of our results, our findings point to novel classification possibilities of graphs at the node level as well as to the information-theoretic classification of the structural complexity of graph indices. Based on these considerations and their underlying mathematical basis, which is the main part of this paper, we want to contribute to the *theoretical* advancement of complex networks—beyond their *empirical* analysis.

The paper is organized as follows. We start with a recollection of some graph-theoretical notions which we use throughout the paper. We speak of the construction (Definition 3) of a graph $G$ from two input graphs $G_1$ and $G_2$ as the so-called joining of these graphs by means of a common node. This operation allows for generating simple connected undirected graphs $G(s, m)$ with $\mathbb{N} \ni s, m \geq 1$ (see Example 1) that will be used in the proof of Theorem 2. In the next section we prove Theorem 1 and Theorem 2. Finally, we discuss the implications of our results (Section Discussion) and draw a conclusion (Section Concluding Remarks).

## Preliminaries

We start with some definitions from graph theory to be used throughout this paper (see [39] for detailed definitions). Let $G$ be a simple connected undirected graph with the *vertex set* $V = V(G)$ and the *edge set* $E = E(G)$. The *order n* of $G$ is the number of its vertices ($n = |V|$). The *size* of $G$ is the number of its edges.

The *degree* $\deg(v)$ of a vertex $v$ of a graph $G$ is the number of edges incident to $v$ in $G$. The *geodesic distance* $\delta(v, w)$ of two vertices $u$ and $v$ in graph $G$ is the number of edges of the shortest path in $G$ connecting them. The diameter $D(G)$ of a graph $G$ is the maximum of geodesic distances in $G$.

By $L(G)$ we denote the average geodesic distance in graph $G = (V, E)$ [1] (where $[X]^2$ is the set of all pairs of elements of the set $X$):

$$L(G) = \frac{\sum_{\{v,w\}\in[V]^2}\delta(v, w)}{n(n-1)}. \tag{1}$$

Further, we denote the numerator of the fraction in (1) by $\Sigma(G)$, that is:

$$\Sigma(G) = \sum_{\{v,w\}\in[V]^2} \delta(v, w). \tag{2}$$

Thus, (1) can be rewritten as:

$$L(G) = \frac{\Sigma(G)}{n(n-1)}. \tag{3}$$

Further, for every vertex $c \in V$ we denote the sum of $n-1$ geodesic distances from $c$ to vertices in $V\backslash\{c\}$ by $\Sigma(c, G)$:

$$\Sigma(c, G) = \sum_{u\in V}\delta(c, u) \tag{4}$$

and using this notation we write:

$$\Sigma(G) = \sum_{u\in V}\Sigma(u, G). \tag{5}$$

The compactness $C_B(G)$ of a graph $G = (V, E)$ with $|V| = n > 1$ was introduced in [31]. In [38] it was observed that for connected graphs the formula for $C_B(G)$ can be represented as follows:

$$C_B(G) = \frac{n}{n-1} - \frac{L(G)}{n-1}. \tag{6}$$

Roughly speaking, for a connected graph $G$ the compactness $C_B(G)$ measures the closeness of all nodes in $G$ to each other. From (6) we see that $C_B(G) \leq 1$.

**Remark 1**. *From* (6) *it follows that if we try to get the limit value of compactness $C_B$ for some sequence of simple connected undirected graphs whenever their order $n$ tends to $+\infty$ we have*

$$\lim_{n\to+\infty} C_B(G) = 1 - \lim_{n\to+\infty}\frac{L(G)}{n-1}. \tag{7}$$

*Thus, for* $\beta = \lim_{n \to +\infty} \frac{L(G)}{n-1}$, *we immediately obtain*

$$\lim_{n \to +\infty} C_B(G) = 1 - \beta.$$

**Definition 1**. *The path graph* $P_m$, $m \geq 2$, *is a simple connected undirected graph with two vertices of degree 1 (called terminal vertices) and* $m - 2$ *vertices of degree 2 (called internal vertices).*

The order $n$ of $P_m$ is equal to $m$ and its diameter $D(P_m) = m - 1$. The vertices of $P_m$ can be labeled by the consecutive integers $\{1, 2, \ldots, m\}$ in such a way that the terminal vertices are labeled by 1 and $m$, respectively, and for every integer $i$, $1 \leq i \leq m - 1$, the consecutive vertices with labels $i$ and $i + 1$ are adjacent.

Further we need the following formulas the proof of which one can easily get in view of (2) and (4) by means of straightforward calculations:

$$\Sigma(P_m) = \frac{m(m^2 - 1)}{3} \tag{8}$$

and

$$\Sigma(c, P_m) = \frac{m(m - 1)}{2}, \tag{9}$$

where $c$ is one of the terminal vertices of $P_m$. So, with (3) and (6) we have

$$C_B(P_m) = \frac{m}{m - 1} - \frac{m + 1}{3(m - 1)} = \frac{2}{3} + \frac{1}{3(m - 1)}. \tag{10}$$

**Definition 2**. *The complete graph* $K_s$ *of order n is a simple undirected graph with s vertices such that each pair of distinct vertices is connected by a unique edge.*

That is, the average geodesic distance $L(K_s)$ equals 1. Obviously, the following equalities hold:

$$\Sigma(K_s) = s(s - 1), \tag{11}$$

$$\Sigma(c, K_s) = s - 1 \tag{12}$$

*where c is an arbitrary vertex in the graph* $K_s$. *Using* (6), *we get the following statement about the compactness* $C_B(K_s)$ *of* $K_s$:

$$C_B(K_s) = \frac{s}{s - 1} - \frac{L(K_s)}{s - 1} = \frac{s}{s - 1} - \frac{1}{s - 1} = 1. \tag{13}$$

It is worth noting that the complete graph $K_s$ is the only graph for which $C_B(K_s)$ equals 1.

**Definition 3**. *Suppose we have two simple undirected connected graphs* $G_1$ *of order* $n_1$ *and* $G_2$ *of order* $n_2$. *We choose a vertex* $c_1$ *in* $G_1$ *and a vertex* $c_2$ *in* $G_2$ *and construct a new graph G by merging* $c_1$ *and* $c_2$ *into the so-called* common vertex $c$ *(that replaces* $c_1$ *and* $c_2$). *We speak of the graph G generated by joining two graphs* $G_1$ *and* $G_2$ *by means of a common vertex c.*

Note that the new graph $G$ is a simple connected undirected graph of order $n = n_1 + n_2 - 1$.

**Example 1**. *Let us take for the graph* $G_1$ *the graph* $K_3$ *(see Definition 2) and for the graph* $G_2$ *the path graph* $P_4$ *(see Definition 1). Then we choose one of the vertices of* $K_3$ *as* $c_1$ *and one of the terminal vertices of* $P_4$ *as* $c_2$. *The graph G is then obtained by joining* $K_3$ *and* $P_4$ *by means of the common vertex c that merges* $c_1$ *and* $c_2$ *as exemplified in* Fig 1.

We denote the graph $G$ by $G(3, 4)$ where the number 3 is the order of $K_3$ and the number 4 is the order of $P_4$. We see that the order of the graph G(3,4) equals to $G = 3 + 4 - 1$.

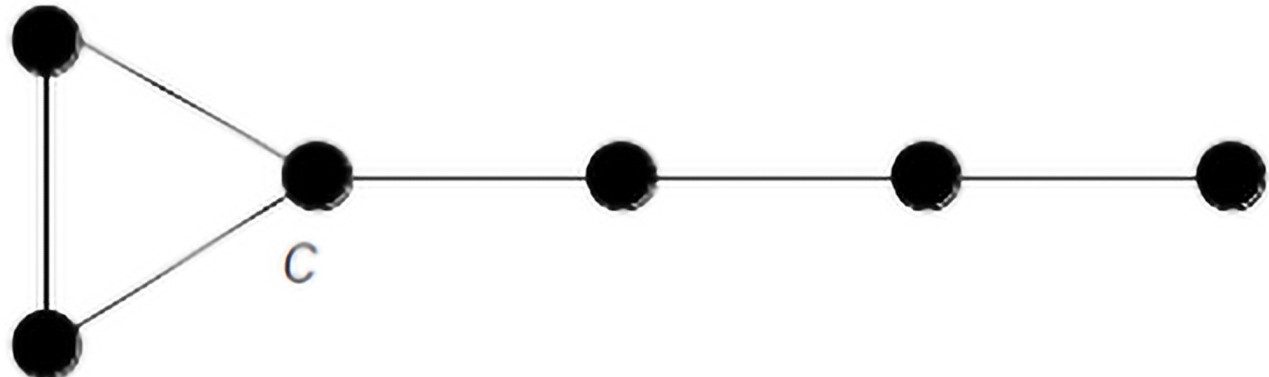

**Fig 1. Graph $G = G(3, 4)$ obtained by joining $G_1 = K_3$ and $G_2 = P_4$ with a common vertex $c$.**

*More generally, we take any vertex $c_1$ of the graph $K_s$ ($s > 1$) and one of the terminal vertices $c_2$ of the graph $P_m$ ($m > 1$) to generate the graph G using the common vertex c (see Definition 3 above). We denote this graph G by $G(s, m)$ also called the fusion graph induced by $K_s$ and $P_m$ with the order n of $G(s, m)$ is $n = s + m - 1$.*

**Proposition 1**. *Let G be the graph that is constructed by joining two simple undirected connected graphs $G_1$ of order $n_1$ and $G_2$ of order $n_2$ with the common vertex c. Then the following holds:*

$$\Sigma(G) = \Sigma(G_1) + \Sigma(G_2) + 2(n_2 - 1)\Sigma(c, G_1) + 2(n_1 - 1)\Sigma(c, G_2). \tag{14}$$

*The proof of this statement is straightforward using formulas (2) and (4).*

**Example 2** *With Proposition 1 we calculate the sum of all geodesic distances $\Sigma(G(s, m))$ in the graph $G(s, m)$, where $G_1 = K_s$, $G_2 = P_m$, $n_1 = s$ and $n_2 = m$. So, from (14) we have:*

$$\Sigma(G(s, m)) = \Sigma(K_s) + \Sigma(P_m) + 2(m - 1)\Sigma(c, K_s) + 2(s - 1)\Sigma(c, P_m)$$

*where $c = \{c_1, c_2\}$ is a common vertex such that $c_1 \in V(K_s)$ and $c_2 \in V(P_s)$. With (11), (12), (8) and (9) we get*

$$\Sigma(G(s, m)) = s(s - 1) + \frac{m(m^2 - 1)}{3} + 2(m - 1)(s - 1) + 2(s - 1)\frac{m(m - 1)}{2} =$$
$$= (s - 1)(s + m - 2 + m^2) + \frac{m(m^2 - 1)}{3}. \tag{15}$$

## Main results

**Theorem 1**. *Given any undirected connected simple graph G of order $n > 2$ which is not isomorphic to the path graph $P_n$ we have $C_B(G) > C_B(P_n)$ or, equivalently, $L(G) < L(P_n)$.*

*Proof.* We note first that if the graph G contains a cycle then we remove one of the edges of this cycle and get another connected undirected simple graph $G'$ of the same order $n$ for which we have

$$L(G') > L(G).$$

So we may assume that G contains no cycle.

Suppose that the diameter $D$ of the graph G is equal to $k < n - 1$ ($k = n - 1$ implies G is isomorphic to the path graph $P_n$). That means that in our graph G there is a simple path (with no

repeating vertices) of length $k$. We enumerate the vertices of this path just as we have done in the previous section (Definition 1). So we have the vertices $A_1, A_2, \ldots, A_{k+1}$ where $A_1$ and $A_{k+1}$ are terminal vertices of our path and the other ones are internal vertices. Clearly the degree of the terminal vertices $A_1$ and $A_{k+1}$ in $G$ is equal to 1 because our path is the largest simple path in the graph $G$. If the vertex $A_2$ has degree 2 we take the next vertex $A_3$ and so on till we get the vertex $A_i$ of degree greater than 2 in the graph $G$. If there is no such vertex in our path then our graph is clearly isomorphic to $P_n$ (recall that $G$ is connected).

So let a vertex $A_i$ be the first vertex of degree greater than 2 in the graph $G$. It means that all vertices $A_2, A_3, \ldots A_{i-1}$ have degree 2 in the graph $G$ and there is a vertex $B$ which is adjacent to the vertex $A_i$ and does not belong to our largest path. Now we remove the edge connecting the vertices $B$ and $A_i$ and join then the vertices $A_1$ and $B$ with an edge.

Thus we get a new graph $G'$ without cycles for which the diameter $D$ is greater that the diameter of the the graph $G$ because the largest simple path of $G'$ contains at least one edge more than the largest simple path of $G$. In view of $i > 1$ we have

$$L(G') > L(G).$$

Further we repeat this procedure with our new graph $G'$ and so on till we get the path graph $P_n$. It will be less than $n - k$ steps and after each step we obtain a new graph for which the diameter (the average geodesic distance $L$) is greater than the diameter (the average geodesic distance), respectively, of the next-to-last graph. Thus, our Theorem is proved.

So, we may say now that the graph $P_n$ is the least compact graph among all the simple connected undirected graphs of the same order $n$. On the contrary, the graph $K_n$ is the most compact graph ($C_B(K_n) = 1$). By means of (10) we see immediately that for any simple connected undirected graph $G$ of order $n$ that is not isomorphic with either $P_n$ or $K_n$, the following holds:

$$\frac{n+1}{3} = L(P_n) > L(G) > L(K_n) = 1$$

or, equivalently,

$$\frac{2}{3} + \frac{1}{3(n-1)} = C_B(P_n) < C_B(G) < C_B(K_n) = 1$$

Hence, the limit value of compactness $C_B(G)$ (of $L(G)/n$) for any sequence of simple connected undirected graphs $\{G\}$ lies in the interval $[2/3;1]$ ($[0;1/3]$), respectively. We show now that any number $\alpha \in [2/3;1]$ ($\beta \in [0;1/3]$) can be a limit value of compactness ($L(G)/n$), respectively, for some graph family.

**Theorem 2**. *For every number $\alpha \in [2/3;1]$ there exists a sequence of simple connected undirected graphs for which the limit value of compactness is exactly $\alpha$.*

*Proof.* We first note that, given Remark 1, it suffices to prove that for any number $\beta$ in the interval $[0;1/3]$ there exists a sequence of simple connected undirected graphs $G$ for which

$$\lim_{n \to +\infty} \frac{L(G)}{n-1} = \beta$$

holds, where $n$ is the order of $G$. To prove this, we consider the graphs $G = G(s, m)$ of order $n = s + m - 1$ defined above. With (3) we have

$$L(G(s, m)) = \frac{\Sigma(G(s, m))}{(s + m - 1)(s + m - 2)}.$$

In Example 2, we expressed $\Sigma(G(s, m))$ by means of (15). Hence,

$$L(G(s, m)) = \frac{(s-1)(s+m+m^2-1) + \dfrac{m(m^2-1)}{3}}{(s+m-1)(s+m-2)} \qquad (16)$$

holds. Now we construct the desired sequence of graphs. For a constant number $p \in (0; +\infty)$ which is not necessary rational we set $s = [pm]$ where $[pm]$ is an integer part of the number $pm$, or equivalently, $s = pm - \epsilon_m$ with $\epsilon_m$ ($0 \le \epsilon_m < 1$) being a fraction part of the number $pm$. In the case that $s = [pm] \in \{0, 1\}$, we set $s = 2$ (this can happen if $m$ is not large enough).

So the sequence of graphs $G = G(pm - \epsilon_m, m)$, $m = 2, 3, \ldots$, is well defined and we have with recollection that the order $n$ of graph $G = G(pm - \epsilon_m, m)$ is $n = pm - \epsilon_m + m - 1$ the following:

$$\lim_{m\to\infty} \frac{L(G(pm - \epsilon_m, m)}{pm - \epsilon_m + m - 2}$$

$$= \lim_{m\to\infty} \frac{(pm - \epsilon_m - 1)(pm - \epsilon_m + m + m^2 - 1) + \dfrac{m(m^2-1)}{3}}{(pm - \epsilon_m + m - 2)^2(pm - \epsilon_m + m - 1)}$$

$$= \lim_{m\to+\infty} \frac{\cancel{m^3}\left(\left(p - \dfrac{\epsilon_m}{m} - \dfrac{1}{m}\right)\left(\dfrac{p}{m} - \dfrac{\epsilon_m}{m^2} + \dfrac{1}{m} - \dfrac{1}{m^2} + 1\right) + \dfrac{1}{3} - \dfrac{1}{3m^2}\right)}{\cancel{m^3}\left(1 + p - \dfrac{\epsilon_m}{m} - \dfrac{1}{m}\right)\left(1 + p - \dfrac{\epsilon_m}{m} - \dfrac{2}{m}\right)^2}$$

$$= \frac{p + \dfrac{1}{3}}{(1 + p)^3}.$$

This means that for each constant $p > 0$ and the corresponding sequence of graphs $G = G$ $(pm - \epsilon_m, m)$ with the order $n = pm - \epsilon_m + m - 1$ it holds that:

$$\lim_{n\to+\infty} \frac{L(G)}{n - 1} = \frac{p + \dfrac{1}{3}}{(1 + p)^3}$$

We see that the function $y = f(p) = \frac{p+\frac{1}{3}}{(1+p)^3}$ is defined for all $p \ge 0$, has the negative derivative $f'(p) = \frac{-2p}{(1+p)^4}$ for $p > 0$ and $f'(0) = 0$. Hence, $f(p)$ is monotonously decreasing for $p \ge 0$ and has its maximum which is $\frac{1}{3}$ at the point $p = 0$.

So the plot of the function $y = f(p)$ can be represented as shown in Fig 2. We note that the function $y = f(p)$ maps one-to-one the interval $[0;+\infty)$ onto the interval $\left(0; \frac{1}{3}\right]$ which implies that for every number $\beta \in \left(0; \frac{1}{3}\right]$ there is the only number $p_\beta$ such that for the corresponding sequence of graphs $G = G([p_\beta m], m)$ the following holds:

$$\lim_{m\to\infty} \frac{L(G([p_\beta m], m))}{[p_\beta m] + m - 2} = \beta. \qquad (17)$$

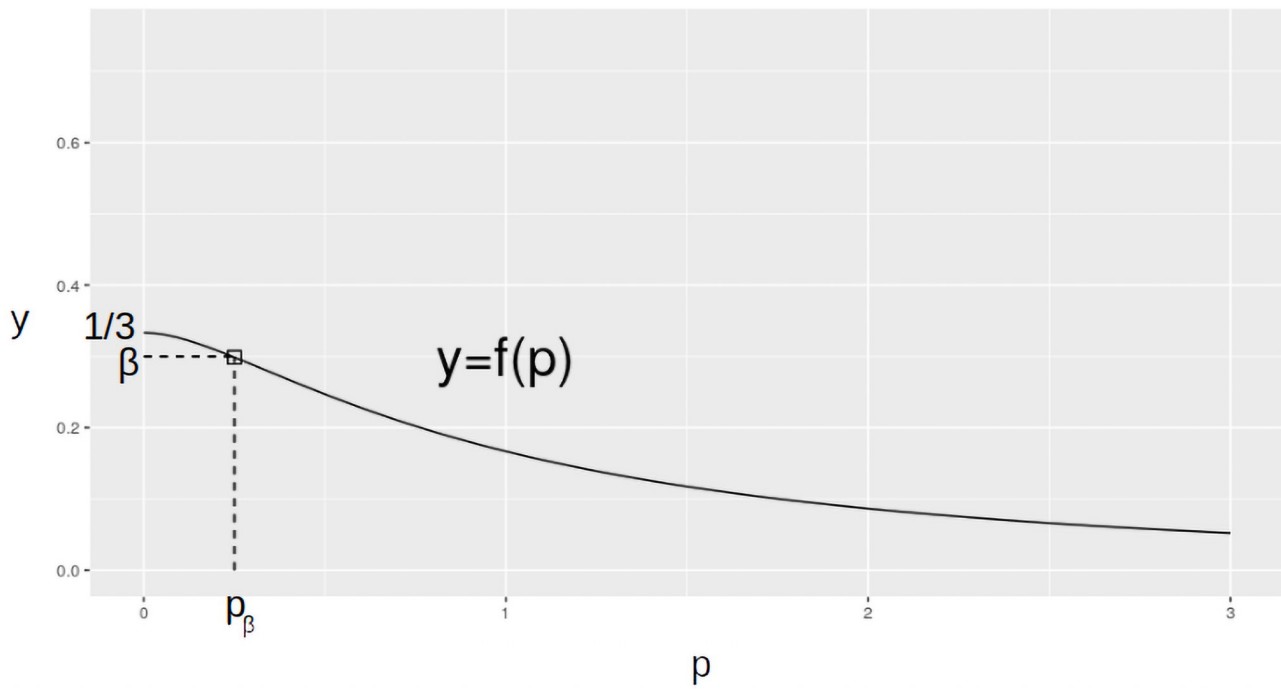

**Fig 2. Plot of the function *y* = *f*(*p*).**

## Discussion

As mentioned in the introduction, the average geodesic distance $L$ is an important graph index in many applications of complex network theory to real-world problems. Consequently, our starting point to analyze the compactness measure was formula (6) which describes the compactness of a connected graph as a function of $L$. Using this measure, one obtains an estimate for the distance structure of a graph that includes all node pairs even in disconnected graphs, given a penalty factor for disconnected nodes, as done in [2]. That is, while $L$ does not change when adding nodes to a graph without linking them by considering only its largest connected component, $C_B$ actually decreases under the perspective of considering all vertices. Moreover, if one adds a connected graph $G'$ to a graph $G$ without connecting any of its vertices to any of the vertices in $G$, there are $V(G') \times V(G)$ many additional pairs of vertices that contribute non-zero values to the compactness of the resulting graph. This makes compactness a candidate for studies of modular, possibly disconnected networks. In [40], for example, the authors examine the graph representations of the semantic memories of persons with Asperger syndrome, which reveal a so-called hyper-modular lexicon structure different from that of the control group. The compactness measure here opens a perspective on distance structure analysis beyond $L$ in particular when the modules are disconnected.

Regardless of the definition of $C_B$ as a function of $L$ for connected graphs, compactness is an informative measure that captures more structural information than $L$ alone in the case of disconnected graphs: it points a way to extend the latter to cover both connected and disconnected graphs. This motivates applying $C_B$ wherever $L$ is already applied, as mentioned in the introduction: a previously untapped opportunity for applications of the compactness measure in complex network theory. However, by our research, we also know that $\frac{2}{3}$ is a lower bound for the compactness of a connected graph. This raises the question of graph indices which, while referring to $L$, capture the disconnectedness of nodes by exhausting the *entire* unit interval.

Notwithstanding the latter research perspective, our results indicate a particular relationship between values above this lower bound. More specifically, according to Theorem 2 we have shown that for any simple connected graph $G$ of order $n$ for which $L(G) = \beta(n-1)$, there exists a sequence of fusion graphs

$$\mathcal{S} = G([p_\beta 2], 2), \ldots, G([p_\beta m], m)$$

derived from a sequence of pairs of graphs

$$\mathcal{P} = (K_{[p_\beta 2]}, P_2), \ldots, (K_{[p_\beta m]}, P_m)$$

such that Eq (17) holds. In this way, the geodesic structure of $G$ as represented by $L$ is reproduced by the latter sequence $\mathcal{S}$ of fusion graphs. In other words, in the limit, a certain sequence of graphs, each arising from the fusion of a fully connected graph and a path graph, is isomorphic to $G$ *with respect to L*. From this point of view, several research questions arise, two in a narrower sense and one in a broader sense, the latter offering a broader perspective for extending our approach:

1. In a narrower perspective, we can consider the sequence $\mathcal{P}$ as a kind of decomposition where the sequence $K_{[p_\beta 2]}, \ldots, K_{[p_\beta m]}$ is associated with $G$'s subnetwork of dense neighborhoods, while the sequence $P_2, \ldots, P_m$ is associated with $G$'s subnetwork of remote neighborhoods: the smaller $p_\beta$, the smaller the order of the graphs $K_{[p_\beta m]}$ in $\mathcal{P}$, the higher the average distances between the nodes in $G$ and vice versa. Moreover, there must be a smallest $n'$ associated with an $m'$ such that $n \leq n' = [p_\beta m'] + m' - 1$, henceforth called division number of $G$ and denoted by $\mathcal{N}(G) = n'$. This division number gives rise to further interpretations. For example, in a labeled variant of $G$, nodes at shorter average distances from each other can then be assigned to $K_{[p_\beta m']}$ (henceforth called *dense nodes*), while those with larger distances are lined up in $P_{m'}$ (so-called *remote nodes*). In this way, the nodes of $G$ are partitioned into two sets according to the decomposition $\mathcal{S}$ and $m'$ (except for the vertex of $G$ ($[p_\beta m'], m'$) by which the component graphs are fused). From this perspective, our approach to proving Theorem 2 is related to a special kind of *graph clustering* approach which distinguishes two classes of vertices. Let us take this perspective one step further: dense and remote vertices can be specified according to their contribution to the compactness of a graph. Deleting remote vertices will hardly influence this compactness, while deleting the former will have higher impact. From this point of view, we gain access to information about the stability of networks—as an alternative to information provided, for example, by centrality measures (e.g. betweenness centrality—[41]). In the same line of thought, adding a node to $G$ causes it to be classified as either dense or remote. This perspective (on the addition or deletion of nodes and their effects on information flow) in turn bridges to the theory of percolation in complex networks [42]: for a set of nodes $V' = \{v_{n+1}, \ldots, v_{n+k}\}$ added stepwise to a graph $G$ of order $n$ with division number $n'$, we finally obtain a series of division numbers $\mathcal{N} = (n+1)', \ldots, (n+k)'$, which characterize their classification into dense and distant nodes depending on the graphs $G_1, \ldots, G_k$ resulting from these additions. The series $\mathcal{N}$ can then be viewed as characterizing the effect of adding and linking the nodes from $V'$ to $G$ and may finally be used for comparing the impacts of processing different vertex sets $V', V'', \ldots$

2. Our results show that given a number $\alpha \in [2/3; 1]$, we can construct a graph series with this number as the compactness of this series in the limit. In other words, for compactness as

the focal graph index and its associated range of values, we induce the existence of a graph series that approximates any desired compactness value in that range. This consideration leads to the idea of creating graph series also in the context of other graph indices by focusing on specific elements from their value ranges. In this sense, our research opens a perspective for a way of looking at graphs that starts from desirable values of graph invariants to generate graphs with these values.

3. Under Theorem 2, $L$ appears as a graph index that does *not* distinguish $G$ and the sequence $\mathcal{S}$. And because of the functional dependence of compactness on $L$ studied here, the same assessment holds analogously for $C_B$. Thus, to regain this distinguishability, we can ask in a further perspective about graph indices other than $L$, to what extent they can be represented by decompositions of the kind considered here. More precisely, we can ask for graph indices $\iota$ that *cannot* be reconstructed by sequences of type $\mathcal{S}$ but require, for example, more than just two graphs per fusion, graphs which are at the same time more complex than fully connected graphs and path graphs. We may ask, for example, how to reproduce measures of graph entropy as considered by [43]. Obviously, the more such graphs are required per fusion and the more complex these component graphs, the more complex the reproduction of $\iota$. This kind of complexity may be an indicator of the expressiveness or informativeness of $\iota$ with respect to the structure of networks. It is probably not reasonable to try to extend this complexity to the highest possible number of vertices required for the fusion of the corresponding component graphs. However, we can still ask for sufficiently complex graph indices that require more than two component classes of sufficient complexity each for the corresponding decomposition. In this way, our approach to proving Theorem 2 can also be seen as naturally related to the qualification of graph indices regarding their structural information value: the more graph classes are needed for the kind of reconstruction exemplified by Theorem 2 and the higher their algorithmic complexity in the sense of [44], the higher the structural information value of that index. At the same time, our approach of functionally linking graph indices (and thus, for example, reconstructing $C_B$ as a function of $L$), gives reason to relate such qualifications to classes of such indices.

These considerations open different perspectives on the theoretical and, to some extent, the future practical relevance of the relationship between average geodetic distance and compactness, for which we have laid a foundation in this paper. This concerns the classification of nodes, the effects of their deletion and insertion on this classification, the construction of graph series to study specific elements of the range of values of graph indices, and the evaluation of the information complexity of such indices. In future work, we aim to elaborate these research perspectives.

## Concluding remarks

In this article we proved that for each sequence of simple connected undirected graphs the limit value of compactness and of the average geodesic distance divided by $n$, $n \to \infty$, lies in the interval [2/3;1] and [0;1/3], respectively. Second, we proved that for every number $\alpha \in [2/3;1]$ ($\beta \in [0;1/3]$) one can construct a sequence of simple connected undirected graphs for which the limit value of compactness ($L(G)/n$) is exactly $\alpha$ ($\beta$). It is worth noting that in order to construct such a sequence we need only two (classes of) graphs $P_m$ (the least compact) and $K_s$ (the most compact) connecting them by means of a common vertex and varying the parameters $s$ and $m$ accordingly. In this way, we arrive at the interpretation that the structural information value of compactness can be reconstructed by instances of two elementary graph classes. On the one hand, this insight reduces the interpretation load associated with observing

certain compactness values in real networks, since we consider all complex connected graphs from the perspective of the fusion of two simpler graph-like building blocks. On the other hand, our research opens a perspective on the reconstruction of related measures (e.g., centrality measures) by analogy with compactness. This leads to a new way of characterizing graph indices according to their structural information value, or the structural complexity they represent, by asking for the simplest possible graph classes that allow for the desired decomposition into graph series. The reason is that the fewer graph classes are needed for such a reconstruction and the lower their algorithmic complexity (cf. Chaitin:1987), the lower the structural information value of the corresponding index. To deepen this research perspective in future work, we will extend our approach to other graph indices to determine the component graph classes associated with them in the latter sense and the graph series based thereon. This will be done not only for the purpose of deriving the corresponding index values under the limit perspective, but also for the purpose of determining the structural complexity of these indices.

## Acknowledgments

This work has been funded by German Federal Ministry of Education (BMBF) in the framework of the research project *Linguistic Networks: Text Technological Representation, Computational Linguistic Synthesis and Physical Modeling*. Financial support by the BMBF and the DFG is gratefully acknowledged.

## Author Contributions

**Conceptualization:** Tatiana Lokot, Olga Abramov, Alexander Mehler.

**Formal analysis:** Tatiana Lokot.

**Funding acquisition:** Alexander Mehler.

**Methodology:** Tatiana Lokot, Olga Abramov.

**Project administration:** Alexander Mehler.

**Validation:** Olga Abramov.

**Writing – original draft:** Tatiana Lokot, Olga Abramov.

**Writing – review & editing:** Tatiana Lokot, Olga Abramov, Alexander Mehler.

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
