## [Decision Letter · Decision Letter 0]

8 Jul 2021

PONE-D-21-17697

About the possible limit values of compactness of some graph classes

PLOS ONE

Dear Dr. Abramov,

Thank you for submitting your manuscript to PLOS ONE. After careful consideration, we feel that it has merit but does not fully meet PLOS ONE’s publication criteria as it currently stands. Therefore, we invite you to submit a revised version of the manuscript that addresses the points raised during the review process.

The manuscript should be revised carefully, the authors should put a considerable effort to improve it by taking into account the comments of the two referees, who raised important remarks related to the related work, methodology, introduction and concluding sections.

Usage of the English and presentation should be improved as well. Please take carefully into account the comments all the two referees for improving the manuscript.

We look forward to receiving your revised manuscript.

Kind regards,

Sergio Consoli

Academic Editor

PLOS ONE

Journal Requirements:

Reviewers' comments:

Reviewer's Responses to Questions

**Comments to the Author**

1. Is the manuscript technically sound, and do the data support the conclusions?

Reviewer #1: Yes

Reviewer #2: Yes

2. Has the statistical analysis been performed appropriately and rigorously? 

Reviewer #1: N/A

Reviewer #2: N/A

3. Have the authors made all data underlying the findings in their manuscript fully available?

Reviewer #1: Yes

Reviewer #2: Yes

4. Is the manuscript presented in an intelligible fashion and written in standard English?

Reviewer #1: Yes

Reviewer #2: Yes

5. Review Comments to the Author

Reviewer #1: The authors give proofs for two theorems that they already stated in their previous paper from 2018, also published in PLOS ONE. The first theorem gives 2/3 and 1 as the lower and upper bounds for the range of limit values corresponding to the compactness metric for any sequence of simple connected graphs. The second theorem states that one can construct the sequence of graphs for which the limit value of compactness falls within these bounds. While the paper is technically sound and very well written, I would like to touch upon some issues.

Major issues:

- The paper provides proofs to two theorems that were stated (without proof) in another paper, published three years ago. I would like to remark that, although the manuscript has been submitted as a stand-alone paper, it would have been perhaps suitable as an appendix to the 2018 publication. A significant proportion of the preliminary materials from the 2018 publication are repeated in the current submission.

- On a similar note, the authors claim in the “Concluding remarks” section that they will discuss the case of unconnected graphs in a follow-up paper. It is up to the editor to decide if the current results should be grouped with the future results into a more substantial publication or if instead, they should be separated as the authors propose.

- In the introduction, the authors mention that the results provided in the manuscript are relevant to “answer questions regarding the structure of hypermedia”. Then, the authors move directly to the technical aspects of the paper. In my opinion, the paper would benefit from a more exhaustive introduction to help the reader understand the value and the context of the results.

- Similarly, I would suggest to the authors to give a short review in the introduction that links their work with the existing literature on compactness.

- Including the expression “of some graph classes” in the title seems vague given that the authors study only one particular class of graphs. A more concrete title would be: “About the possible limit values of compactness of simple connected undirected graphs”.

Minor issues:

- In page 2, the authors give basic graph-theoretic definitions for the building-blocks of the paper. To remain coherent, they should also include what they mean by “simple graph”, “connected graph”, “sequence of graphs”, “isomorphic graphs”, “undirected graph”, “adjacent vertices”.

- In page 3, the authors introduce the mathematical definition of compactness. The intuition behind this concept can be deducted after reading the whole manuscript. However, the paper would be more readable if this definition was accompanied by the corresponding intuition of what this metric is measuring.

- Regarding the writing style, the authors use phrases as “obviously, …” or “one can easily see that …”. It is recommended to avoid these expressions as some statements may not be at all obvious for some readers (perhaps they only started researching graphs, or they may not have a mathematical background).

Reviewer #2: The reviewed article main contributions are theorem 1 and 2, regarding the compactness measure of graphs. In the first theorem, it is shown that the compactness value in the case of undirected connected simple graphs lies within a defined interval, while the second one deals with sequences of graphs for a given compactness value. Moreover, a perspective for the given results is given, along with some instrumental definitions and examples.

The methodology of the article and the claimed results seem sound. They justify opting for publication. However, some aspects would need to be considered.

1) While the authors reference some relevant bibliography, the wider context within which the article is situated is not successfully described. It would be desirable to provide the reader with clear information about what the contributions of the article are within the current state of affairs in the field. In particular, only one of the cited publications is from 2018 and is from the same author. The rest are at least from 2015 or older. In general, a broader perspective in terms of the offered references would be, to the reviewer criteria, needed. Along with this, it is recommended when the citations are made in the text, to provide the reader with the summarized information and the specific aspects for which the citation is made.

2) The importance and relevance of the produced work should be highlighted, which links to point 3.

3) In general, while the structure of the text is suitable, the abstract, the introduction, as well as the conclusions sections would need some improvement.  

Some suggestions are:

In the case of the abstract, to include the context and the relevance of the study. For the introduction, to present a wider context for the article, and probably leaving mathematical expressions for further in the text. Concluding remarks could also be improved including a summary of some of the information in the text, namely, the perspectives on future research offered; with a reference to relevance again; and to how the authors have contributed to the described current research context.

4) Finally, although it seems that the authors have successfully built upon their work Lokot et al. 2018. The similarity of the title and how it is referred to in the introduction do not help in making it clear.

Some minor aspects:

In line 126, implies that (instead of than).

In line 133, brackets are missing around 17 to be consistent with the rest of the text.

6. PLOS authors have the option to publish the peer review history of their article (what does this mean?). If published, this will include your full peer review and any attached files.

Reviewer #1: No

Reviewer #2: No

---

## [Author Response · Author response to Decision Letter 0]

9 Oct 2021

We provide all answers to reviewer comments in a separate file "response to reviewers".

---

## [Decision Letter · Decision Letter 1]

27 Oct 2021

On the asymptotic behavior of the average geodesic distance

L and the compactness C_B of simple connected undirected

graphs whose order approaches infinity

PONE-D-21-17697R1

Dear Dr. Abramov,

We’re pleased to inform you that your manuscript has been judged scientifically suitable for publication and will be formally accepted for publication once it meets all outstanding technical requirements.

Kind regards,

Sergio Consoli

Academic Editor

PLOS ONE

Additional Editor Comments (optional):

Reviewers' comments:

Reviewer's Responses to Questions

**Comments to the Author**

1. If the authors have adequately addressed your comments raised in a previous round of review and you feel that this manuscript is now acceptable for publication, you may indicate that here to bypass the “Comments to the Author” section, enter your conflict of interest statement in the “Confidential to Editor” section, and submit your "Accept" recommendation.

Reviewer #1: All comments have been addressed

Reviewer #2: All comments have been addressed

2. Is the manuscript technically sound, and do the data support the conclusions?

Reviewer #1: Yes

Reviewer #2: (No Response)

3. Has the statistical analysis been performed appropriately and rigorously? 

Reviewer #1: Yes

Reviewer #2: (No Response)

4. Have the authors made all data underlying the findings in their manuscript fully available?

Reviewer #1: Yes

Reviewer #2: (No Response)

5. Is the manuscript presented in an intelligible fashion and written in standard English?

Reviewer #1: Yes

Reviewer #2: (No Response)

6. Review Comments to the Author

Reviewer #1: I would like to congratulate the authors for successfully addressing the comments. The inclusion of the additional references improves the article significantly. Similarly, the explanation for the context of their research and for the intuition behind the concept of compactness, make the article more readable.

There are some minor suggestions for the authors:

- Line 32: does not make full grammatical sense, did you mean 'additionally' instead of 'additional'?

- Line 185: did you mean 'giving' instead of 'given'?

Reviewer #2: (No Response)

7. PLOS authors have the option to publish the peer review history of their article (what does this mean?). If published, this will include your full peer review and any attached files.

Reviewer #1: No

Reviewer #2: No

---

## [Editor Report · Acceptance letter]

2 Nov 2021

PONE-D-21-17697R1 

On the asymptotic behavior of the average geodesic distance *L* and the compactness *C*_B_ of simple connected undirected graphs whose order approaches infinity 

Dear Dr. Abramov:

I'm pleased to inform you that your manuscript has been deemed suitable for publication in PLOS ONE. Congratulations! Your manuscript is now with our production department. 

Kind regards, 

on behalf of

Dr. Sergio Consoli 

Academic Editor

PLOS ONE